# Devil is in Details: Locality-Aware 3D Abdominal CT Volume Generation for Self-Supervised Organ Segmentation

## ABSTRACT

In the realm of medical image analysis, self-supervised learning techniques (SSL) have emerged to alleviate labeling demands, while still facing the challenge of training data scarcity owing to escalating resource requirements and privacy constraints. Numerous efforts employ generative models to generate high-fidelity, unlabeled 3D volumes across diverse modalities and anatomical regions. However, the intricate and indistinguishable anatomical structures within the abdomen pose a unique challenge to abdominal CT volume generation compared to other anatomical regions. To address the overlooked challenge, we introduce the Locality-Aware Diffusion (Lad), a novel method tailored for exquisite 3D abdominal CT volume generation. We design a locality loss to refine crucial anatomical regions and devise a condition extractor to integrate abdominal priori into generation, thereby enabling the generation of large quantities of high-quality abdominal CT volumes essential for SSL tasks without the need for additional data such as labels or radiology reports. Volumes generated through our method demonstrate remarkable fidelity in reproducing abdominal structures, achieving a decrease in FID score from 0.0034 to 0.0002 on AbdomenCT-1K dataset, closely mirroring authentic data and surpassing current methods. Extensive experiments demonstrate the effectiveness of our method in self-supervised organ segmentation tasks, resulting in an improvement in mean Dice scores on two abdominal datasets effectively. These results underscore the potential of synthetic data to advance self-supervised learning in medical image analysis.

## CCS CONCEPTS

• **Applied computing** → **Imaging**; • **Computing methodologies** → **3D imaging**.

## KEYWORDS

3D image generation, Medical imaging, Conditional image generation, Image reconstruction

## 1 INTRODUCTION

Due to the difficulty of acquiring and hand-labeling large amounts of volumetric medical data, an increasing share of medical models are trained with self-supervised learning (SSL) techniques [16, 18, 24, 33], which exploits complex structures in large-scale unlabeled data to enhance efficiency and effectiveness [16]. However,

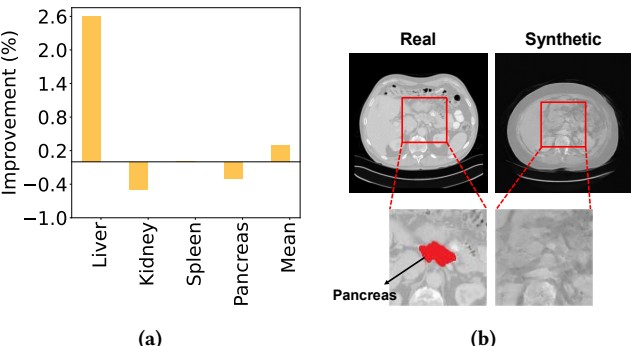

**(a)**           **(b)**

**Figure 1: Our Motivation. (a) Improvements in Dice score for the self-supervised segmentation model SSL-ALPNet [22] trained on the augmented AbdomenCT-1K dataset (which augmented with abdominal CT images generated by Medical Diffusion [14]) over those trained on the original AbdomenCT-1K dataset [20]. Despite the synthetic data augmentation, there is a notable decrease in SSL performance across 20% augmentation rate, indicating the limited utility of sub-optimal synthetic abdominal CT data. (b) Comparison of real and synthetic abdominal CT images. The outline of the pancreas in the real image is much more discriminative than in the synthetic image generated by Medical Diffusion [14].**

high-quality unlabeled data for training encounters the obstacle of scarcity. Medical volume acquisition requires significant resource investments [19, 25, 37]. Furthermore, the sensitive nature of medical images exacerbates these complexities, necessitating careful consideration of privacy preservation measures [28, 34].

To save the costs on data acquisition, unlabelled synthetic data is used to enhance SSL tasks in a cost-effective and practical way [21, 27]. Unlabelled medical volumes are usually generated by Generative Adversarial Networks (GANs) [7] or diffusion models [11]. Noteworthy studies [14, 29] present versatile generation methods succeeding in high-resolution 3D volumes of various modalities and anatomical regions, showing promising prospects for synthetic images in downstream medical research. However, challenges such as unstable training and mode collapse problems in GAN hindered the generation of large-scale, diverse, realistic volumes [13, 29, 38]. Due to the superior image generation capabilities of diffusion models, they are widely used in creating synthetic data for medical applications. These synthetic datasets often integrated into downstream tasks by either substituting real data in the training set or augmenting small training sets. However, while diffusion models [14] excel in generating realistic volumes for various modalities and anatomical regions, synthetic abdominal CT volumes fall short of expectations. Directly integrating synthetic abdominal CT volumes into downstream models will lead to performance degradation. As shown in Figure 1a, the dice scores of the self-supervised

segmentation model SSL-ALPNet [22] trained on the augmented AbdomenCT-1K dataset, which is augmented with 20% of synthetic abdominal CT images generated by Medical Diffusion [14], achieve an improvement of mean dice scores, but are mostly and unexpectedly worse on small organs such as pancreas and spleen than those trained on the original AbdomenCT-1K dataset. Therefore, it is urgent to investigate how to generate high-quality synthetic volumes of abdominal CT with realistic anatomical structures by the diffusion model.

Compared to other anatomical regions, the abdomen presents a unique challenge due to its intricate and indistinguishable anatomical structures. This complexity makes the synthesis of high-fidelity 3D abdominal CT volumes particularly challenging. Synthetic abdominal CT images frequently lack the detailed anatomical structures required for clear delineation, particularly evident in blurred contours of small organs such as the pancreas. For example, Figure 1b illustrates an abdominal CT image generated by Medical Diffusion [14], showcasing distorted and blurred anatomical details compared to real images. For example, the pancreas is challenging to distinguish in synthetic abdominal CT images. This phenomenon is attributed to generative models operating solely from a global perspective, neglecting the refinement of intricate details. Consequently, synthetic abdominal CT images with unrealistic anatomical structures may cause the model to learn biased visual representations in SSL tasks [21, 27], resulting in inferior performance.

To address the above issue, we present a locality-aware 3D abdominal CT volume generation method named **L**ocality-**A**ware **D**iffusion(Lad), focusing on detailed organ-specific information instead of the entire image during generation. Our work comprises three phases: Latent Space Construction, Diffusion Fitting in Latent Space, and Sampling in Latent space. During the Latent Space Construction phase and Diffusion Fitting in Latent Space phase, we construct the latent space for CT image using VQ-GAN [5] to focus on abdominal structure and train a diffusion probabilistic models [11] to fit into the latent space under the guidance of the image content and structure. Specifically, to localize the regions of anatomical structures, we use **Priori Extraction** module to predict the mask of abdominal organ, like the pancreas, to localize the regions of anatomical structures. The predicted mask denotes the locality we focus on to refine and serves as a valuable anatomical priori in our whole generation process. To construct a latent space that fully captures the features of the original data, we designed the **Locality Refinement** module. This module focuses the reconstruction model on a sub-volume cropped according to the predicted mask, enhancing the fidelity of abdominal structure reconstruction and refining the quality of the latent space. When fitting Diffusion into the latent space, to extract locality information effectively, we introduce the **Locality Condition Extractor** $E_c$, which captures both content and structural perspectives from the predicted mask. The mutually complementary combination of content and structure information guides the diffusion model to learn the distribution of latent vectors better. For sampling in the latent space, we apply **Locality Condition Augmentation** module to expand the original maskset. Subsequently, Condition Extractor $E_c$ extracts abundant conditions from the augmented maskset, guiding the generation of massive volumes.

Our synthetic data achieves the best scores across all metrics (FID [10], MMD [8], MS-SSIM [30]) for synthesis quality in quantitative comparisons and has the closest distribution to real data in qualitative comparisons on two datasets, AbdomenCT-1K [20] dataset and TotalSegmentator [31] dataset, demonstrating realism in both holistic and localized regions of abdominal CT volumes. By experimenting with treating synthetic data as real data and augmentation methods, respectively, we effectively improve the performance of the self-supervised segmentation model SSL-ALPNet [22] on the AbdomenCT-1K dataset, especially bringing performance gains in small abdominal organs like pancreas and spleen.

The main contribution of our work is threefold:

(1) We pioneer **L**ocality-**A**ware **D**iffusion(Lad), the first method tailored for exquisite 3D abdominal CT volume generation. With a dedicated focus on locality details in the generation process, Lad produces abdominal CT images with more delicate anatomical structures, which is crucial for self-supervised learning to extract in-distribution representations.

(2) We employ locality refinement, locality condition extraction, and locality condition augmentation, respectively, to enhance the reconstruction, fitting, and sampling of CT images within the latent space, with heightened focus on local anatomical structures.

(3) Our method can generate large amounts of high-quality abdominal CT volumes, which prove highly effective in SSL organ segmentation tasks without requiring additional data such as labels or radiology reports. This underscores the viability of our synthetic volumes as highly effective alternatives for self-supervised learning.

## 2 RELATED WORK

### 2.1 3D Medical Image Generation

The proliferation of generative models in natural images has spurred the development of numerous methods dedicated to medical image generation. Notable contributions by Peng *et al.* [23], Yoon *et al.* [35], and Shibata *et al.* [28] have showcased the synthesis of high-fidelity 3D brain MRI using diffusion models. Additionally, Sun *et al.* [29] and Khader *et al.* [14] contributed versatile generation methods capable of producing high-resolution 3D volumes across diverse modalities and anatomical regions, such as thoracic CT, knee MRI, and brain MRI. These advancements hold promise for the integration of synthetic images into subsequent medical investigations. Differing from these prior investigations, our focus lies in the realm of 3D Abdominal CT volume generation. We confront the challenge posed by intricate, localized anatomical structures, aiming to address this critical gap in the field.

### 2.2 Image Generation from Conditions

A number of papers infusing condition information into the generation process in recent years are closely related to our work. The recent ControlNet [36] incorporates fine-tuned spatial conditioning controls to Stable Diffusion [26], a large pre-trained text-to-image latent diffusion model. Unlike ControlNet relying on first encoding an input mask to some latent space before feeding it to the diffusion model, we additionally extract topological structure features from masks to integrate more anatomical structure information.

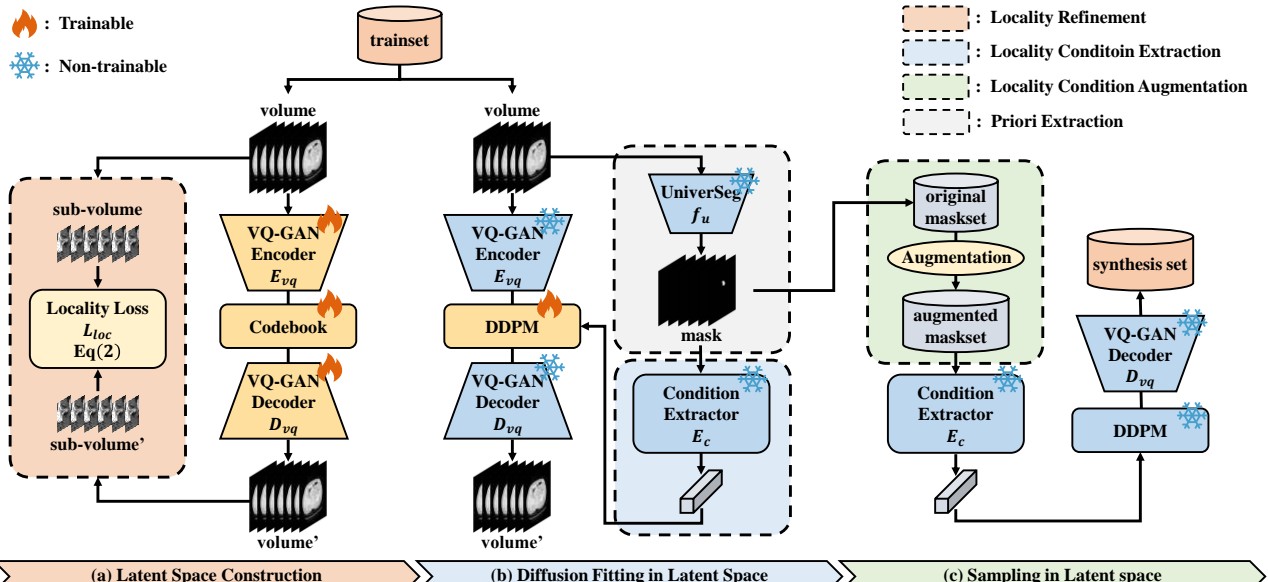

**Figure 2: Overview of Locality-Aware Diffusion (Lad).** Lad consists of three phases: (a) Latent space construction. We introduce a locality refinement module into the VQ-GAN [5] training. Refinement module uses the Locality Loss $\mathcal{L}_{loc}$ to facilitate VQ-GAN to learn more details of anatomical structures in the low-dimensional space. (b) Diffusion fitting in latent space. We introduce a locality condition extraction module into the diffusion [11] training. The diffusion model incorporates a locality condition extracted by Condition Extractor $E_c$ to generate volumes. (c) Sampling in latent space. We introduce a locality condition augmentation module into the diffusion sampling. In the augmentation module, diverse conditions are extracted from the augmented maskset and used to generate massive volumes. All three phases use the masks output by a priori extraction module. Masks are predicted by a well-trained universal segmentation model UniverSeg, with masks considered as priori.

There are studies incorporating additional medical data to aid in generation tasks. Segmentation-Guided Diffusion [15] adds segmentation guidance to diffusion models by concatenating the mask channel-wise to the network input and incorporates a random mask ablation training algorithm to enable synthesis flexibility. Unlike Segmentation-Guided Diffusion needing additional mask labels, we leverage predictions of organs from a well-trained universal segmentation model [1], used alongside unlabeled data and serving as priori knowledge to indicate regions to refine and generate high-quality large amounts of samples valuable to SSL tasks. Moreover, Xu *et al.* [32] and Hamamci *et al.* [9] have introduced innovative methods for text-guided volumetric generation, leveraging 3D chest CT scans paired with radiology reports. Unlike these methods, our method does not necessitate integrating text or data from other modalities, thereby alleviating the need for additional resources.

## 3 METHOD

### 3.1 Overview

Lad comprises three phases: Latent Space Construction, Diffusion Fitting in Latent Space, and Sampling in Latent space.

As a crucial thread running through the entire method, predicted mask of abdominal organ like the pancreas from a well-trained universal segmentation model, UniverSeg [1] $f_u$, localizes the regions of anatomical structures and serves as an anatomical priori where we can dig into abdominal details. We denote the predicted mask of the $i_{th}$ abdominal CT volume $x^i$, as $y^i$, $i = 0, 1, ..., |D_{tr}|$, $D_{tr}$ means the trainset of abdominal CT volumes.

Central to the efficacy of this method is the utilization of a predicted abdominal organ mask generated by a well-trained universal segmentation model, UniverSeg [1] denoted as $f_u$. This mask serves as an anatomical prior, facilitating the localization of anatomical structures within the abdominal region. Specifically, denoting the predicted mask for the $i_{th}$ abdominal CT volume $x^i$ as $y^i$, where $i = 0, 1, ..., |D_{tr}|$ and $D_{tr}$ represents the training set of abdominal CT volumes. The precision of the predicted mask $y^i$ is insignificant. In scenarios where the prediction aligns closely with ground truth, detailed features of the abdominal organ can be effectively extracted by the mask. Conversely, inaccuracies in the prediction, indicating misalignment with the ground truth, present challenges for the segmentation model in identifying the correct anatomical regions. Therefore, enhancing precision in these areas is crucial for providing discriminative information.

### 3.2 Locality Refinement

Considering computational requirements, the diffusion model is trained and samples in the latent space of vector quantized autoencoders (VQ-GAN) [5]. So, producing high-quality reconstructions is crucial to the quality of images generated by a diffusion model modeling the latent space. We first define a loss function $\mathcal{L}_{glo}$ of VQ-GAN following [14]:

Given the computational demands, the diffusion model operates within the latent space of vector quantized autoencoders (VQ-GAN) [5]. The quality of the latent space directly impacts the fidelity of images generated by the diffusion model. To ensure the latent

space fully captures the features of the original data, we define a loss function $\mathcal{L}_{glo}$ for VQ-GAN, following [14]:

$$\mathcal{L}_{glo} = \mathcal{L}_{rec} + \mathcal{L}_{codebook} + \mathcal{L}_{commit} + \mathcal{L}_{perc} + \mathcal{L}_{match} + \mathcal{L}_{disc} \quad (1)$$

where $\mathcal{L}_{rec}$ is a reconstruction loss, $\mathcal{L}_{codebook}$ is a codebook loss, $\mathcal{L}_{commit}$ is a commitment loss, $\mathcal{L}_{perc}$ is a perceptual loss, $\mathcal{L}_{match}$ is a feature matching loss [6] and $\mathcal{L}_{disc}$ is a discriminator loss.

Apparently, all of these loss functions prioritize global reconstruction quality, ignoring the importance of detail reconstruction quality, particularly in anatomical structures. The significance of detail reconstruction varies across regions, where accurate depiction of anatomical structures holds paramount importance. Even if background reconstruction is excellent, blurred details of anatomical structures significantly hinder subsequent generation tasks. Thanks to the predicted mask $y^i$, we can localize the specific areas within volume $x^i$ that require attention. By cropping volume $x^i$ into a sub-volume $x^i_{sub}$ and performing the same operation on the reconstructed volume $\widehat{x^i}$, we obtain $\widehat{x^i_{sub}}$. Introducing the Globality Loss $\mathcal{L}_{glo}$, defined by Eq (1), we now introduce the Locality Loss $\mathcal{L}_{loc}$. This loss function is tailored to direct the reconstruction model's focus towards the sub-volume $x^i_{sub}$ extracted from volume $x^i$ and the corresponding reconstructed sub-volume $\widehat{x^i_{sub}}$, leveraging the predicted mask $y^i$ to enhance the detail reconstruction of abdominal structures and thereby improve the quality of the latent space. The Locality Loss is formulated as the $L_1$ distance between the sub-volume $x^i_{sub}$ of the input data $x^i$ and the corresponding sub-volume $\widehat{x^i_{sub}}$ of the output data $\widehat{x^i}$:

$$\mathcal{L}_{loc} = \|(x^i_{sub}, \widehat{x^i_{sub}})\|_1, \quad (2)$$

so, the overall objective of our reconstruction model is to minimize the loss function $\mathcal{L}$:

$$\mathcal{L} = \mathcal{L}_{glo} + \lambda \mathcal{L}_{loc}, \quad (3)$$

where $\lambda$ is a hyperparameter that plays a crucial role in balancing between globality and locality, thus enabling fine-grained generation for comprehensive holistic reconstruction. In our experimental setup, we empirically set $\lambda$ to 1.0.

### 3.3 Diffusion Model with Locality Condition

*3.3.1 Conditional Denoising Diffusion Probabilistic Model.* We first encode volume $x^i$ into a low-dimensional latent space through VQ-GAN [5] and subsequently train a denoising diffusion probabilistic model (DDPM) [11] on the latent representation of the volume, denoted as $E_{vq}(x^i)$. To generate anatomical details more precisely, we incorporate the priori knowledge from mask $y^i$ as condition information to guide the generation, instead of generating in-distribution volumes randomly. To be specific, we design Condition Extractor $E_{con}$ to extract locality details as the condition signal $c$ to train the model to fit $p(E_{vq}(x)|c)$, which means the distribution of the latent space of abdominal CT volumes given condition $c$.

*3.3.2 Locality Condition Extraction.* In order to fully incorporate anatomical structure details from priori knowledge, we introduce Condition Extractor $E_c$ to extract locality information from the predicted mask $y^i$ into a condition vector $c^i$ from both content and

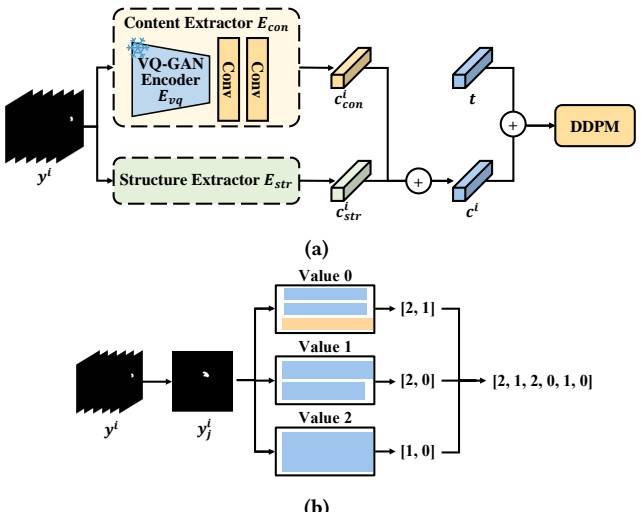

(a)

(b)

**Figure 3: Locality Condition Extraction. (a) Condition Extractor $E_c$. This module extracts anatomical structure details from priori knowledge to guide the generation more precisely, in the joint action of two complementary sub-modules, i.e., Content Extractor $E_{con}$ and Structure Extractor $E_{str}$. (b) The specific mechanism of Structure Extractor $E_{str}$. Structure Extractor $E_{str}$ extracts topological features of the slice $y^i_j$ of mask $y^i$ into a one-dimension vector of length 6 according to the Betti numbers of each label value.**

structure perspectives, as Figure 3a. The combination of content and structure information enables the diffusion model to learn the distribution of latent vectors better.

To entail condition $c^i$ with abdominal features, we first use the encoder of the former trained VQ-GAN $E_{vq}$ to encode mask $y^i$ into the latent space of abdominal CT volume $x^i$ and then pass it through two convolution layers in order to save computation memory. We call this extraction process Content Extractor $E_{con}$ and denote the output as content vector $c^i_{con}$. $c^i_{con}$ contains image-level information such as the size, shape and location of the abdominal organ. Since $E_{vq}$ is trained on abdominal CT volume trainset $D_{tr}$, content vector $c^i_{con}$ represents encoding mask $y^i$ into an image-level latent space, without making the most of the structure information of mask label. Besides, content vector $c^i_{con}$ represents the global features of the entire mask. Apparently, the significance of the background cannot match up to the foreground's.

Considering the above reasons, we additionally extract the organ's structure information to fully utilize the organ itself. Structure Extractor $E_{str}$ is devised to extract topological features of the mask $y^i$ by using cubical complex to represent the $j$th slice $y^i_j$ of $y^i \in \mathbb{R}^{D \times H \times W}$, $j = 0, 1, ..., D-1$ and then computing *Betti numbers* of each cubical complex. The Betti numbers, $\beta_k$ counts the number of features of dimension $k$, where $\beta_0$ is the number of connected components, $\beta_1$ the number of loops or holes, $\beta_2$ the number of hollow voids, *etc* [4]. Considering the common organ's topological structure, only the first three Betti numbers is considered in our method, as Figure 3b. Thus, there are three labels in slice $y^i_j$, the value of each label is 0 representing the background, 1 representing

the organ, and 2 representing tumors. We create different cubical complexes for each label. So the final structure information of $y^i_j$ is represented as $\{\beta_{k,l}\}$, which is a one-dimension vector has 6 elements, $l$ meaning label value $\in \{0, 1, 2\}$. As for the whole volume $y^i$, we concatenate the topological features of all its slices and denoted as $c^i_{str}$.

The two extracted condition vectors, named content condition $c^i_{con}$ and structure condition $c^i_{str}$ individually, are mutually complementary and concatenated to represent the abdominal features, which act as condition vectors to guide the generation.

## 3.4 Sampling With Locality Condition Augmentation

*3.4.1 Locality Condition Augmentation.* To generate large amounts of volumes, we apply common data augmentation techniques, including flipping, random affine, and random elastic deformation, to expand the original maskset predicted by UniverSeg [1]. We use the augmented maskset as the abundant condition source to be extracted by Condition Extractor $E_c$ for generation guidance.

*3.4.2 Sampling with Condition Guidance.* We follow *classifier-free guidance* [12] to guide the diffusion model so that the denoise model adjusts predictions $\widetilde{\epsilon_\theta}$ constructed via

$$\widetilde{\epsilon_\theta}(z_t, c) = (1 + w)\epsilon_\theta(z_t, c) - w\epsilon_\theta(z_t), \qquad (4)$$

where $z_t \sim q(z_t|x^i)$ means the input of the denoise model, which can be computed by adding Gaussian noise to $E_{vq}(x^i)$ at timestep $t \in \{0, 1, ..., T - 1\}$. $w$ is the guidance strength of condition $c$. $\epsilon_\theta(z_t, c)$ is the regular conditional model prediction, and $\epsilon_\theta(z_t)$ is a prediction from an unconditional model jointly trained with the conditional model by randomly setting $c$ to the unconditional class identifier $\emptyset$ with probability $p_{uncond}$. To balance the quality and diversity, we set $p_{uncond} = 0.25$ in training and set $w = 1.0$ in subsequent sampling. See more details about the experiments on parameter $w$ in Section 4.4.2.

## 4 EXPERIMENTS

## 4.1 Datasets and Experimental Settings

*4.1.1 Datasets.* To examine the robustness and generalizability of our method, We use two public 3D datasets with different distributions for 3D abdominal CT image generation in the experiments:

*AbdomenCT-1K.* AbdomenCT-1K Dataset [20] collects 1,112 high-resolution 3D abdominal CT images from 12 medical centers, 1000 of which have manual annotations of four abdominal organs, including the liver, kidney, spleen, and pancreas. Considering the following tests of the utilization of synthetic data in downstream tasks, 800 CT images without masks are randomly selected for the training phase and sampling phase in the generation process, while the other 200 CT images are reserved for the application tests with their corresponding ground truth mask.

*TotalSegmentator.* TotalSegmentator [31] provides 1,228 CT volumes covering 117 classes annotated by voxel, encompassing information on over 20 abdominal organs. The size of these slices ranges from 47 to 499, and some of these volumes cover limited areas of the entire abdomen. To ensure the integrity of abdomen synthesis,

we charge off these incomplete volumes. Moreover, considering the downstream tests, volumes, where the liver, kidney, spleen, and pancreas don't exist, are discarded, too. In the end, 767 volumes are used in our experiments. 80% (614 volumes) of these are used as the training set and 20% (153 volumes) are used as the testing set.

*4.1.2 Pre-processing.* In order to enhance data utility, we apply resampling techniques to both datasets, adjusting voxel spacing to [1.6, 1.6, 2.3] and [1.1, 1.1, 1.5], respectively. Following resampling, we standardize the height and width dimensions to 256. To make the most of the pancreas masks, we employ a strategic cropping method. Utilizing a sliding window mechanism with a window size of 32, we traverse the entire volume along the Z-axis, aligning with ground truth annotations. Consequently, the processed data dimensions are uniformized to $256 \times 256 \times 32$. Furthermore, to ensure uniformity in image intensity, we truncate voxel values to the range of $[-1000, 400]$ and subsequently normalize them to the interval $[0, 1]$.

*4.1.3 Implementation Details.* We train the VQ-GAN for 100,000 steps. We set the compression rate as $(4, 4, 4)$. We let the learning rate be $3 \times 10^{-4}$. The batch size is set as 2. Then, we train the diffusion model for 150,000 steps with 300 timesteps, the learning rate of $1 \times 10^{-4}$, and the batch size of 20. All experiments are performed on two NVIDIA A100 GPUs.

## 4.2 Quality evaluation of synthetic data

We thoroughly evaluated the quality of synthetic data using 800 synthetic volumes from AbdomenCT-1K dataset and 610 synthetic volumes from TotalSegmentator dataset, matching the size of the training set.

*4.2.1 Quantitative Comparison.*

*Synthetic Data with Most Realistic Volumes.* We quantitatively evaluate the realism of synthetic volumes using Fréchet Inception Distance (FID) [10] and Maximum Mean Discrepancy (MMD) [8]. Lower FID/MMD values indicate closer distributions of synthetic volumes to real ones, implying more realistic synthetic volumes. To evaluate the ability to synthesize intricate details, we additionally cropped sub-volumes from the synthesis set based on the maximum bounding box of the abdominal organ in the original maskset, calculating localized FID and localized MMD for these sub-volumes. Due to HA-GAN being trained only with volumes of size $128^3$ or $256^3$, we resized the depth of volumes in the trainset from 32 to 256 and resized them back after sampling. For computing FID and MMD, we utilized a 3D ResNet model pre-trained on 3D medical images [3] to extract features, following [29]. As shown in Table 1, HA-GAN [29] exhibits limitations in capturing the distribution of abdominal CT volumes despite its proficiency in generating high-resolution 3D thorax CT and brain MRI scans. On the other hand, Medical Diffusion [14] achieves superior performance with lower FID and MMD scores across both datasets, showcasing the robust generative capabilities inherent in diffusion models. Notably, Lad outperforms the aforementioned methods across two abdominal datasets, excelling both holistically and in localized evaluations. Particularly impressive are its FID and MMD scores, which plummet to 0.0002 and 0.0003, respectively. These results underscore

**Table 1: Quantitative Comparison of Synthetic Volumes on AbdomenCT-1K and TotalSegmentator Datasets. The best scores in each column are highlighted in bold. FID and MMD metrics assess the realism of synthetic volumes generated by different methods, while MS-SSIM evaluates diversity. Additionally, localized FID/MMD scores are calculated for sub-volumes cropped from the synthesis set based on the maximum bounding box of the abdominal organ in the original mask set. Our method, Lad, achieves the highest scores in these metrics from both holistic and localized perspectives, demonstrating the significant contribution of our attention to detail to the overall enhancement of volume quality.**

| Method | AbdomenCT-1K [20] | | | | | TotalSegmentator [31] | | | | |
| | FID↓ | | MMD↓ | | MS-SSIM↓ | FID↓ | | MMD↓ | | MS-SSIM↓ |
| | Holistic | Localized | Holistic | Localized | | Holistic | Localized | Holistic | Localized | |
| Real | — | — | — | — | 0.5719 | — | — | — | — | 0.4447 |
| HA-GAN [29] | 0.4958 | 0.0302 | 1.4138 | 0.3685 | 0.9992 | 0.2889 | 0.1696 | 1.1055 | 0.7922 | 0.9987 |
| Medical Diffusion [14] | 0.0034 | 0.0005 | 0.0149 | 0.0075 | 0.6085 | 0.0012 | **0.0005** | 0.0031 | **0.0003** | 0.4584 |
| Lad (Ours) | **0.0002** | **0.0002** | **0.0003** | **0.0011** | **0.5940** | **0.0007** | **0.0005** | **0.0015** | 0.0011 | **0.4574** |

the efficacy of our method in generating high-fidelity 3D abdominal CT volumes. The meticulous attention to anatomical nuances significantly contributes to the enhancement of overall volume quality.

*Synthetic Data with Most Diverse Volumes.* We assess the diversity of each method using the multi-scale structural similarity metric (MS-SSIM) [30]. MS-SSIM is computed by averaging the results of 400 synthetic sample pairs within each method, serving as a representation of the MS-SSIM within the internal synthesis set. Higher MS-SSIM scores imply that the synthetic volumes generated by a method are more alike, whereas lower scores signify increased diversity. As illustrated in Table 1, HA-GAN encounter mode collapse, achieving super high MS-SSIM score in dealing with abdominal CT volumes. On the contrary, Medical Diffusion ensures diversity while maintaining generation quality. Lad attains the lowest MS-SSIM score across both datasets, indicating its capability to produce a broader range of samples that faithfully represent the original data distribution. In addition to enhancing the quality of synthetic samples, Lad simultaneously achieves an increase in diversity.

### 4.2.2 Qualitative Comparison.

*Qualitative Comparison of Anatomical Structures.* To qualitatively assess the diversity and authenticity of synthetic volumes, Figure 5 presents synthetic samples from each method along with zoomed-in regions on both the AbdomenCT-1K and TotalSegmentator datasets. HA-GAN appears incapable of generating even rough outlines of the abdomen. While Medical Diffusion succeeds in generating complete abdominal structures, the synthesized anatomical details are ambiguous. In contrast, Lad demonstrates superior performance by generating clearer anatomical structures, resulting in more realistic abdominal details.

*Qualitative Comparison of Synthetic Data Distribution.* We embed both synthetic and real volumes into a latent space to assess the degree of overlap in their data distributions. Following the method outlined in [3, 29], we utilize a pre-trained 3D medical ResNet model [17] to extract features from 800 real and synthetic data samples. Subsequently, Multidimensional Scaling (MDS) [2] is employed to map the extracted features into a 2-dimensional space for both the AbdomenCT-1K and TotalSegmentator datasets. For each method, we fit an ellipse to the embedding with the least squares. From Figure 4, it is evident that the data distribution of

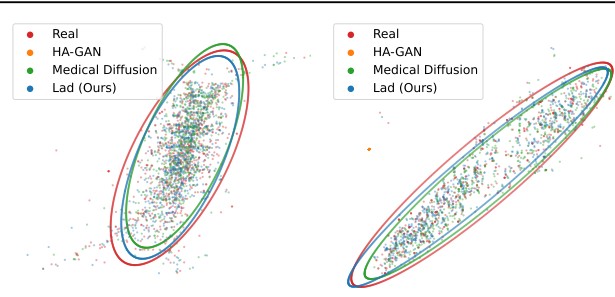

(a) On AbdomenCT-1K          (b) On TotalSegmentator

**Figure 4: Comparison of Synthetic Volumes Embedding From Different Methods on Two Abdominal Datasets. Features extracted from Synthetic volumes are embedded into a 2-dimensional space using MDS, with ellipses fitted to method-specific scatter plots for improved clarity. Both (a) and (b) show that the embedding of Lad exhibits the highest overlap with real volumes.**

synthetic volumes generated by Lad exhibits the highest degree of overlap with real data. This observation suggests that Lad generates volumes with a more realistic appearance compared to other methods.

## 4.3 Synthetic Volumes for Self-Supervised Organ Segmentation

To assess the effectiveness of synthetic data in self-supervised learning (SSL) tasks, we sample 800 synthetic data from both Medical Diffusion [14] and Lad, which are then utilized as the training set for training the self-supervised segmentation model SSL-ALPNet [22]. For each training set, we conduct five training runs of the segmentation model and evaluate its performance, taking the average of the test results. The dice scores obtained in the segmentation tests of models trained on different training sets are presented in Figure 6.

*4.3.1 Synthetic Data for Training.* We employ synthetic data as substitutes for authentic data in training the segmentation model to evaluate the genuine impact of synthetic data on downstream feature learning tasks, without incorporating any real data in the process. As depicted in Figure 6a, the segmentation model trained on synthetic volumes from Lad outperforms the one trained on synthetic volumes from Medical Diffusion in mean dice scores of

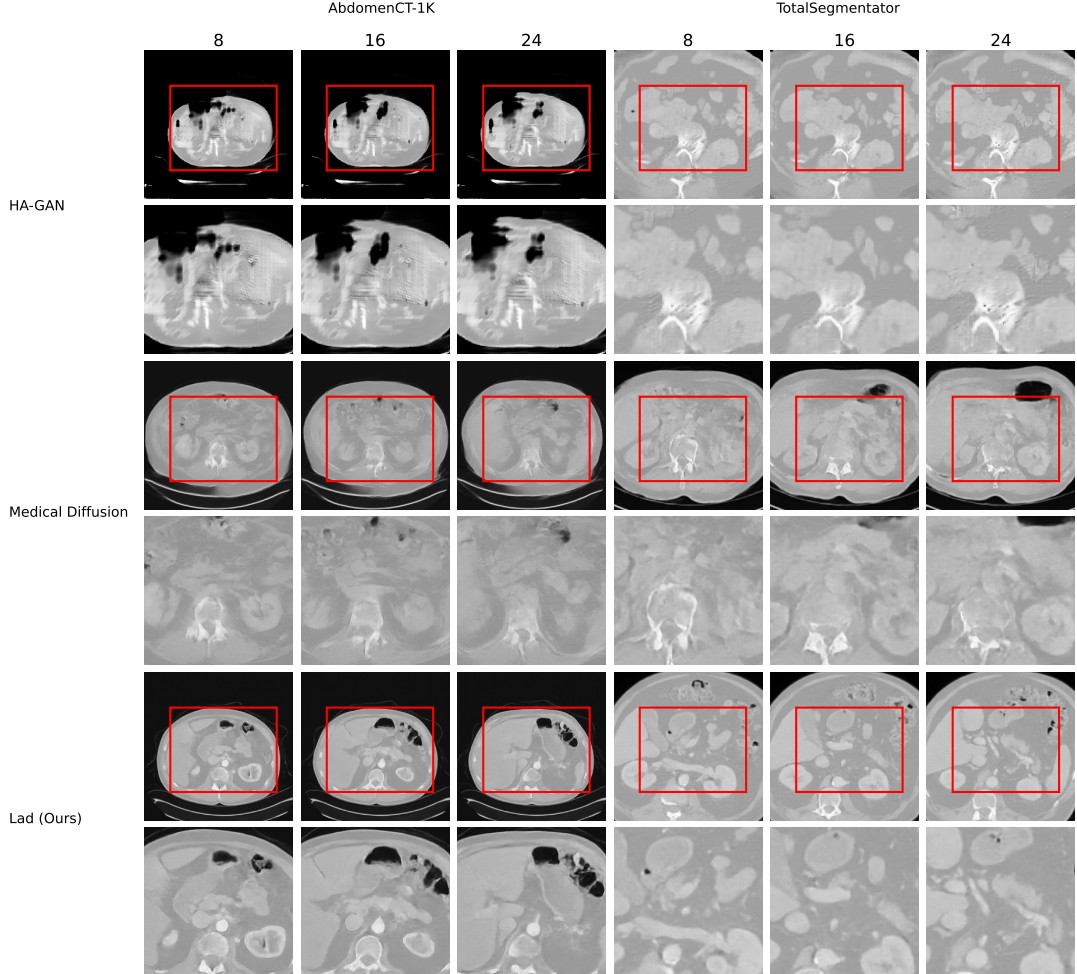

**Figure 5: Visualization of Synthetic Volumes Generated by Different Methods on Abdomenct-1K Dataset and Totalsegmentator Dataset. The first three columns display synthetic samples on AbdomenCT-1K dataset, while the last three columns display samples on TotalSegmentator dataset. Synthetic samples of each method are presented in two rows: the first row depicts the entire image, while the second row focuses on a local area of the image. Additionally, for each 3D volume, slices corresponding to the 8th, 16th, and 24th positions are displayed. This figure highlights the ability of Lad to produce clearer anatomical structures compared to other methods.**

four abdominal organs. As evidenced by higher dice scores for small organs such as the pancreas and spleen, exceeding Medical Diffusion by 0.9% and 2.7% respectively, our method's emphasis on granularity significantly enhances the delineation of anatomical structures' details. Despite prioritizing local features, our approach achieves comparable performance to Medical Diffusion on larger organs such as the liver and kidney.

*4.3.2 Synthetic Data for Augmentation.* We incorporate synthetic volumes, which are approximately 20% the size of the training set, as a form of data augmentation within the self-supervised segmentation model. Figure 6b illustrates that our approach yields the highest mean dice scores, thereby facilitating effective learning of visual representations by the SSL model. Notably, our method demonstrates particular strength in addressing the challenges associated with small organs, effectively bridging the performance gap observed when compared to Medical Diffusion.

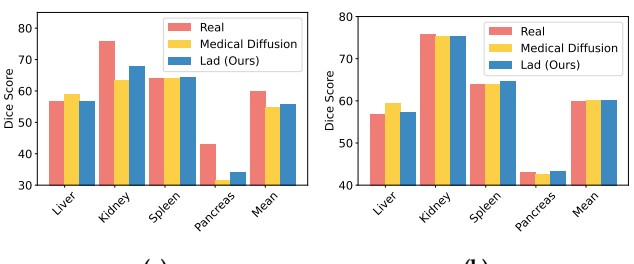

**Figure 6: Abdominal Organ Segmentation Performance Comparison on Different Train sets.**

## 4.4 Ablation and Parameter Studies

*4.4.1 Ablation Studies.* To comprehensively assess the efficacy of each component in producing high-fidelity abdominal CT volumes, our evaluation centers on three key elements of our approach: Locality Loss $\mathcal{L}_{loc}$, Content Extractor $E_{con}$, and Structure Extractor

**Table 2: Ablation Study for Synthetic Volumes Conditioned on Original Mask and Augmented Mask. We validate the function of three parts of our method: Locality Loss $\mathcal{L}_{loc}$, Content Extractor $E_{con}$, and Structure Extractor $E_{str}$. We conduct ablation studies on synthetic volumes conditioned on augmented mask sets to discern the individual impact of each component on synthesis quality. Besides, we extend our analysis to include ablation studies on synthetic volumes conditioned on augmented mask sets without original masks, to gauge the scalability and adaptability of our method in facing "unseen" masks.**

| Original Masks | Version | $\mathcal{L}_{loc}$ | $E_{con}$ | $E_{str}$ | FID↓ | | MMD↓ | | MS-SSIM↓ |
|---|---|---|---|---|---|---|---|---|---|
| | | | | | Holistic | Localized | Holistic | Localized | |
| — | V0 | | | | 0.0034 | 0.0005 | 0.0149 | 0.0075 | 0.6085 |
| ✓ | V1 | | ✓ | | 0.0020 | 0.0004 | 0.0080 | 0.0051 | 0.5984 |
| | V2 | | | ✓ | 0.0002 | 0.0003 | 0.0009 | 0.0020 | 0.5989 |
| | V3 | | ✓ | ✓ | 0.0003 | 0.0004 | 0.0027 | 0.0028 | 0.5983 |
| | Full | ✓ | ✓ | ✓ | 0.0002 | 0.0002 | 0.0003 | 0.0011 | 0.5940 |
| ✗ | V1 | | ✓ | | 0.0027 | 0.0009 | 0.0093 | 0.0078 | 0.5973 |
| | V2 | | | ✓ | 0.0008 | 0.0011 | 0.0067 | 0.0086 | 0.6058 |
| | V3 | | ✓ | ✓ | 0.0006 | 0.0009 | 0.0061 | 0.0070 | 0.6053 |
| | Full | ✓ | ✓ | ✓ | 0.0005 | 0.0004 | 0.0036 | 0.0019 | 0.6051 |

**Table 3: Parameter Study of Guidance Strength in Sampling.**

| $w$ | FID↓ | | MMD↓ | | MS-SSIM↓ |
|---|---|---|---|---|---|
| | Holistic | Localized | Holistic | Localized | |
| 0.5 | 0.0010 | 0.0005 | 0.0035 | 0.0037 | 0.6134 |
| 1.0 | **0.0005** | 0.0005 | **0.0005** | 0.0023 | **0.6002** |
| 1.5 | 0.0006 | 0.0005 | 0.0020 | 0.0031 | 0.6016 |
| 2.0 | 0.0006 | **0.0004** | 0.0010 | **0.0020** | 0.6010 |

$E_{str}$. Initially, we conduct ablation studies on synthetic volumes conditioned on augmented mask sets to discern the individual impact of each component on synthesis quality. Furthermore, to gauge the scalability and adaptability of our method in generating diverse volumes, we extend our analysis to include ablation studies on synthetic volumes conditioned on augmented mask sets without original masks. This scenario simulates encounters with a wide array of "unseen" masks. Table 2 presents a quantitative assessment of synthesized samples across various iterations of our method.

*Locality Attention Enhances Holistic Quality.* A comparison between versions V3 and the Full version, as presented in Table 2, underscores the significant enhancement in synthesized sample quality upon the introduction of Locality Loss $\mathcal{L}_{loc}$. The samples generated by the Full version consistently achieve the highest scores in terms of both realism and diversity, regardless of the presence of original masks. These findings underscore the pivotal role of our attention mechanism towards locality during the generation process. This attention mechanism enriches synthesized samples with intricate anatomical structure details, ultimately elevating the overall quality of abdominal CT volumes.

*Sufficient Condition Extraction Guides Efficient Sample Generation.* A comparative analysis among versions V0, V1, and V2 in Table 2 with original masks reveals the importance of incorporating condition guidance in producing volumes that closely resemble real counterparts, as evidenced by the drop in all metrics. Furthermore, comparing versions V1, V2, and V3, it becomes evident that synthesizing samples with superior scores across all metrics is achievable only when both the Content Extractor $E_{con}$ and Structure Extractor $E_{str}$ are introduced concurrently. This observation underscores

the effectiveness of mutually complementary content conditions ($c^i_{con}$) and structure conditions ($c^i_{str}$). However, upon analyzing the results of ablation studies conducted without original masks, it is intriguing to note that V1, featuring the Content Extractor ($E_{con}$) alone, outperforms V3 in terms of MS-SSIM. This observation is reasonable, as the condition in V3 exerts a more potent control over locality details, potentially leading to decreased diversity, which is a trade-off phenomenon. Moreover, the slightly higher MS-SSIM observed in V2 compared to other versions can be attributed to the similarity in topological structures across different volumes, which imposes a less stringent constraint on volume generation.

*4.4.2 Parameter Study.* The guidance strength $w$ in Eq (4) during sampling represents a trade-off between quality and diversity. We sample with 256 augmented masks for different values of $w$, specifically 0.5, 1.0, 1.5, and 2.0, and subsequently conduct a quantitative evaluation on synthetic data. Table 3 demonstrates that synthetic volumes achieve outstanding performance in metrics of both realism and diversity only when $w = 1.0$.

## 5 CONCLUSION

We introduce Locality-Aware Diffusion (Lad), a pioneering method designed specifically for the precise generation of 3D abdominal CT volumes. With a focus on capturing intricate anatomical structure details, we leverage prior knowledge from a well-trained segmentation model, UniverSeg. Through the incorporation of locality refinement, locality condition extraction, and locality condition augmentation modules, we significantly enhance the overall quality of the generated volumes by directing attention to finer details. Experimental results demonstrate that the synthetic abdominal CT volumes produced by our method exhibit realism and diversity across various metrics. These findings underscore the efficacy of synthetic data in facilitating self-supervised tasks. Our approach not only advances the state-of-the-art in abdominal CT volume generation but also opens up new avenues for leveraging synthetic data in medical imaging research.

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
