# OpenReview forum: "Devil is in Details: Locality-Aware 3D Abdominal CT Volume Generation for Organ Segmentation"
_acmmm.org/ACMMM/2024/Conference — MM2024 Poster_

### Official Review · Reviewer_kDpt · 2024-05-20

**Rating:** 4
**Confidence:** 2

**Summary:**

To address the generation of some subtle and complex anatomical structures in abdominal CT scans, this paper introduces Locality-Aware Diffusion (Lad), which is claimed to be the first method specifically tailored for exquisite 3D abdominal CT volume generation. The paper designs a locality loss to refine key anatomical regions and devises a condition extractor to integrate abdominal priors into the generation process, enabling the generation of a large number of high-quality abdominal CT volumes essential for SSL tasks without the need for additional data, such as labels or radiology reports.

**Strengths:**

1. This paper claims to be the first method specifically tailored for exquisite 3D abdominal CT volume generation.

2. The introduction of the motivation is very clear, and the overall method is also well-written and easy to understand.

3. The ablation study is quite comprehensive, with an ablation of each module.

**Limitations:**

1. Are the contents from pages 320-326 repetitive of those from pages 283-289? They seem to convey almost the same meaning.

2. For both comparative experiments and ablation studies, it is best to display the segmentation results to make them clearer, as shown in Figure 1(b).

3. How does this paper maintain spatial consistency between slices for the 3D generation method?

**Suitability:**

2

---

### Official Review · Reviewer_9A63 · 2024-05-24

**Rating:** 3
**Confidence:** 3

**Summary:**

This manuscript presents a novel method called Locality-Aware Diffusion (Lad), tailored for generating high-quality 3D abdominal CT volumes. It addresses the challenges posed by the intricate anatomical structures of the abdomen, which are often poorly represented in synthetic data. The method integrates a locality loss and a condition extractor to enhance the quality and accuracy of generated volumes. Notably, it achieves significant improvements in FID scores, MMD and MS-SSIM on self-supervised organ segmentation tasks, demonstrating the utility of synthetic data in medical image analysis.

**Strengths:**

1.	Innovative Approach: This manuscript introduces an innovative blend of locality-aware techniques to improve the synthetic 3D abdominal CT volumes, which is a significant step forward in the field of medical imaging.
2.	Significant Empirical Improvements: The method substantially improves FID, MMD, and MS-SSIM scores, demonstrating its effectiveness over existing methods.
3.	Detailed Methodological Framework: This manuscript presents a well-structured and comprehensive methodological framework, introducing the use of locality loss and condition extractors that integrate anatomical priors effectively.

**Limitations:**

1.	The authors choose the MS-SSIM to evaluate the diversity of synthetic data. However, MS-SSIM is commonly used to calculate the structural similarity between generated results and a target image when the target is available. The use of this metric in the paper is inappropriate, and the authors need to reconsider its application.
2.	In Figure 4, though the legend indicates that the figure includes four types of data, the absence of the orange ellipse representing HA-GAN calls into question the authenticity of the conclusions.
3.	The visualization results in Figure 5 are visualized using different slices from a single synthetic volume in each of two datasets, which does not demonstrate the diversity of the method proposed in this manuscript.
4.	In Figure 6, the segmentation model trained on the Lad-generated training set shows weaker performance in liver organ segmentation compared to Medical Diffusion, and the paper lacks an explanation for this phenomenon, raising doubts about the rationality and effectiveness of the methods discussed.
5.	In figure 6, there is a lack of evaluation for the segmentation model trained on data synthesized by HA-GAN.
6.	The method in this paper combines various complex models, including VQ-GAN, UniverSeg, DDPM, etc. Does the increased complexity of the models lead to increase computational demands, thereby limiting their usability?

**Suitability:**

1

---

### Official Review · Reviewer_7Rgp · 2024-05-28

**Rating:** 4
**Confidence:** 3

**Summary:**

The paper introduces a novel method for generating high-fidelity 3D abdominal CT volumes using a locality-aware diffusion model (Lad). The method addresses the challenge of generating detailed and realistic anatomical structures in abdominal CT images, which is critical for improving the performance of self-supervised learning (SSL) in medical image analysis. The proposed approach leverages a locality loss and a condition extractor to focus on refining crucial anatomical regions, enabling the generation of high-quality synthetic data that significantly enhances SSL tasks.

**Strengths:**

Novelty: The introduction of the locality-aware diffusion model (Lad) represents a significant advancement in the generation of 3D abdominal CT volumes. The focus on locality details during the generation process is a novel approach that addresses the specific challenges of generating realistic and detailed anatomical structures in abdominal CT images.

Theoretical Approach and Technical Correctness: The paper provides a well-structured and theoretically sound methodology, including the use of VQ-GAN for latent space construction and a locality condition extractor to integrate anatomical priors into the generation process. The inclusion of locality refinement and locality condition augmentation modules further enhances the model's ability to generate high-fidelity synthetic data.

Adequate Evaluation: The extensive experiments and quantitative evaluations demonstrate the effectiveness of the proposed method. The results show significant improvements in metrics such as Fréchet Inception Distance (FID), Maximum Mean Discrepancy (MMD), and multi-scale structural similarity metric (MS-SSIM), indicating the superior quality and diversity of the generated volumes.

Clarity: The paper is well-written and clearly explains the methodology, experimental setup, and results. The use of figures and tables to illustrate the qualitative and quantitative comparisons of the generated volumes with real data and other methods enhances the clarity of the presentation.

Applications: The proposed method has significant implications for self-supervised learning in medical image analysis. The ability to generate high-quality synthetic data without the need for additional labels or radiology reports can reduce the reliance on expensive and time-consuming manual data annotation, thus advancing the field of medical imaging.

**Limitations:**

Complexity of Method: The proposed method involves several components, including VQ-GAN, locality loss, condition extractor, and augmentation modules, which may increase the complexity of implementation and require substantial computational resources.

Generality Across Modalities: While the method shows promising results for abdominal CT volumes, its effectiveness across other anatomical regions and imaging modalities needs further validation. The specificity of the locality-aware approach to the abdomen may limit its generalizability to other medical imaging tasks.

Dependency on Pre-trained Models: The method relies on the use of pre-trained segmentation models (e.g., UniverSeg) for generating anatomical priors. The performance of the proposed method may be influenced by the accuracy and robustness of these pre-trained models.

**Suitability:**

3

---

### Meta-Review · Area_Chair_Z68J · 2024-07-09

**Recommendation:** Accept (Poster)
**Confidence:** 4

**Metareview:**

After rebuttal, the reviewers generally agreed on the contributions of this paper, i.e. domain generalization using synthetic data for medical image analysis. The authors should add additional visualization results on spatial consistency in the final version.